# Dynamic Tensile Behavior of Woven C/SiC Composite: Experiments and Strain-Rate-Dependent Yield Criterion

**DOI:** 10.3390/ma15134679

**Published:** 2022-07-04

**Authors:** Jingwei Yu, Peiwei Zhang, Qiang Chen, Qingguo Fei

**Affiliations:** 1School of Mechanical Engineering, Southeast University, Nanjing 211189, China; jingweiyu@seu.edu.cn (J.Y.); qiang_chen@seu.edu.cn (Q.C.); 2Jiangsu Engineering Research Center of Aerospace Machinery, Southeast University, Nanjing 211189, China

**Keywords:** C/SiC composite, strain rate, yield criterion, interval analysis

## Abstract

This paper studies the yield behavior of a woven carbon-fiber-reinforced silicon-matrix (C/SiC) composite under dynamic tensile loading. Experiments were carried out to obtain the tensile properties of the C/SiC composite at a strain rate range of 2 × 10^−5^/s to 99.4/s. A strain-rate-dependent yield criterion based on the distortional strain energy density theory is established to describe the yield behavior. The interval uncertainty is considered for a more reliable yield prediction. Experimental results show that the yield stress, elastic modulus, and yield strain of the C/SiC composite grow with the increasing strain rate. The failure mode transitions from progressive crack extension to uneven fiber bundle breakage. The predicted results by the yield criterion match well with experimental data. Experimental results are enveloped within the uncertainty level of 45% in the critical distortional energy density, corresponding to an uncertainty of 14% and 11% in the yield stress and yield strain, respectively. With the support of the proposed strain-rate-dependent yield criterion, the yield behavior of the C/SiC composite under dynamic loading conditions can be predicted with reasonable accuracy.

## 1. Introduction

Carbon-fiber-reinforced silicon-matrix (C/SiC) composites are widely used in aerospace structures due to their advantages of low density, high fracture toughness, and high specific strength [1,2,3,4]. Components made of C/SiC composites are inevitably subjected to dynamic loads, such as the impact of bird strikes or runway debris [5]. The high-speed loading may cause unexpected failure of the composite structures. For example, the loss of the space shuttle Columbia in 2003 was attributed to the impact of foam insulation on the leading edge of the wing [6]. Therefore, an adequate understanding of the mechanical properties of C/SiC composites under dynamic loading conditions is of great significance for the safety of composite structures.

Numerous works have studied the fracture strength, toughness, and corresponding failure mechanism of C/SiC composites under quasi-static loading conditions [7,8,9,10]. Nevertheless, the mechanical properties of C/SiC composites under dynamic loading have not been adequately studied. Li et al. [11] studied the dynamic fracture of C/SiC composites by split Hopkinson pressure bar (SHPB) and gas gun loading. The main fracture mechanism involves void collapse and shear damage banding for the out-of-plane direction compression, while the delamination induces fracture for the in-plane direction. Tensile fracture is related to delamination, fiber pullout, and fiber breaking. Chen [12] investigated the dynamic tensile behavior of 2D-C/SiC composites with SHPB. The tensile strength is observed to increase at a high strain rate. Microscope observations show that the fracture surface displays integrated bundle pullout, indicating enhanced in-bundle interfacial strengthening. Hu et al. [13] studied the effect of strain rate on the interlaminar shear behavior of 2D-C/SiC composites. The interlaminar shear strength increases at a shear strain rate range of 10^−3^/s to 200/s. More fiber fracture and residual SiC matrix are observed after dynamic loading. The different damage processes under quasi-static and dynamic loadings are believed to account for the shear strain rate sensitivity. Luan et al. [14] observed that the compression stiffness, fracture strength, and energy absorption of C/SiC composite increase with increasing strain rate in both in-plane and out-of-plane directions. Suo et al. [15] attributed the increase in compressive strength to the increase in the rate of microcrack nucleation and extension. They observed that fiber bundles exhibit better integrality and less in-bundle debonding under dynamic compressive loading. From the above studies, it can be concluded that the microcracks have enough time to nucleate and extend under quasi-static loading conditions, while, for dynamic loading, the applied loading usually lasts for tens of microseconds, leading to differences in the damage mechanism and failure modes. Thus, the C/SiC composites exhibit significant strain rate sensitivity in their mechanical behaviors, including strength, stiffness, absorbed energy, etc.

In structural integrity analysis, it is necessary to establish proper failure criteria for the safety assessment. Numerous failure theories for composites have emerged over the past six decades [16]. The “World-Wide Failure Exercise” (WWFE) concludes the status of composite failure theories and evaluates the maturity of these criteria for predicting the failure behavior of fiber-reinforced composites [17]. However, the strain rate effects have been neglected in most failure theories. Few studies have considered the strain rate effect in the failure of composites. Gurusideswar et al. [18] studied the influence of strain rate on the tensile strength of glass/epoxy composites. They used an empirical non-linear regression function to describe the relationship between the tensile strength and strain rate. Gillespie et al. [19] introduced a strain-rate-dependent function into the Yen and Caiazzo failure criteria to describe the interlaminar shear strength of glass/epoxy composites. This function is in an empirical form that combines linear and logarithmic laws. Similarly, Daniel [20] introduced a linear regression function into the NU-Daniel criteria to describe the effect of strain rate on the failure of carbon/epoxy composites. From the above studies, it is seen that the strain-rate-dependent function introduced in the failure criteria is empirical without physical meaning, leading to the chosen form of the function being different even for the same material. Hence, it is necessary to establish a failure criterion regarding the strain rate with general applicability for composites.

Another challenge regarding the characterization of C/SiC composites is the uncertain mechanical properties caused by heterogeneity. The irregular voids and cracks in the ceramic matrix, the strength of individual fiber bundles, and the interface bonding status between fiber and matrix could lead to a certain degree of uncertainty in the macro-mechanical performance [21]. This uncertainty is unavoidable in current preparation processing [22,23]. Hence, it is meaningful to introduce uncertainty into the safety assessment of C/SiC composites.

The main purpose of this study is to study the dynamic tensile behavior of the C/SiC composite and establish a dynamic yield criterion. In Section 2, tensile tests are carried out on the C/SiC composite under high-speed loading. In Section 3, the strain rate is introduced into the distortional strain energy density theory to establish the dynamic yield criterion. Further, the interval uncertainty is introduced into the dynamic yield criterion in Section 4. In Section 5, the influence of strain rate on the yield behavior of the C/SiC composite is discussed. The dynamic yield criterion is compared with the experimental results. Conclusions are drawn in the Section 6.

## 2. Experiments

### 2.1. Material

Fabric C/SiC composites were supplied by the State Key Laboratory of Solidification Processing at Northwestern Polytechnical University, China. Nippon Toray Corporation provided the T300-3K carbon fiber. The composite was prepared by chemical vapor infiltration of silicon carbide into a fabricated preform of stacks of carbon fiber woven laminates. Before infiltration, the carbon fibers were deposited with a pyrolytic carbon interfacial layer with a thickness of 0.2 μm. The detailed preparation process is illustrated in [24]. Due to the few interlacing points, the satin fabric of the preform better maintained the tensile properties of the yarns compared with plain or twill fabrics. The resultant composite had a fiber volume fraction of 40% and porosity of 15%. The density of the composite was 1.8 g/cm^3^. The composites were machined into a flat dog-bone-shaped specimen as recommended in ASTM C1275-18 [25], as shown in Figure 1. The thickness of the specimen was 6.2 mm.

### 2.2. Test Procedure

An MTS Criterion^®^ C45 universal electronic test machine with a loading capacity of 50 kN was used to determine the quasi-static properties of the C/SiC composites. The environment temperature and humidity were nearly 20 °C and 40%, respectively. Tensile tests were conducted at a crosshead speed of 0.01 mm/s, corresponding to a strain rate of 2 × 10^−5^/s. Aluminum alloy plates were pasted at the ends of the specimen to prevent unexpected failure in grab sections. The specimens were loaded at a constant speed until fracture.

The dynamic tensile tests were conducted in the Key Laboratory of Impact and Safety Engineering, Ningbo University. A Zwick/Roell HTM 5020 servo-hydraulic high-speed test machine was used to carry out the tests. The test system had a tensile capacity of 50 kN and a maximum loading speed of 20 m/s. The strain rate of experiments was determined by the loading speed controlled by the testing apparatus. The corresponding true strains and strain rates were measured by strain gauges directly bonded to the central zone of specimens. The loading apparatus and specimen are shown in Figure 2.

## 3. Strain-Rate-Dependent Yield Criterion

Various constitutive models have been proposed to describe the influence of strain rate on the relationship between stress and strain. One approach to assess the effect of strain rate is to use a regression function, given as [26]
(1)$$\sigma =\phi \varepsilon +\psi {\dot{\varepsilon }}^{n}$$
where *ϕ*, *ψ*, and *n* are constants determined by experimental data.

For orthotropic materials such as fiber-reinforced ceramic-matrix composites, Equation (1) can be extended as
(2)$$\sigma =C\varepsilon +F{\overset{\cdot }{\varepsilon }}^{n}$$
in which **ε** and $\overset{\cdot }{\varepsilon }$ are the total strain tensor and total strain rate tensor, respectively. **C** denotes the elastic stiffness matrix, and **F** is the coefficient matrix representing the influence of the strain rate component, which is assumed to be similar to the form of elastic stiffness matrix. The symmetric matrix **C** and **F** are given as
(3)$$C=\left[\begin{array}{llllll} \;k_{11} & k_{12} & k_{13} & & & \\ \;k_{21} & k_{22} & k_{23} & & & \\ \;k_{31} & k_{32} & k_{33} & & & \\ & & & k_{44} & & \\ & & & & k_{55} & \\ & & & & & k_{66} \end{array}\right],F=\left[\begin{array}{llllll} \;f_{11} & f_{12} & f_{13} & & & \\ \;f_{21} & f_{22} & f_{23} & & & \\ \;f_{31} & f_{32} & f_{33} & & & \\ & & & f_{44} & & \\ & & & & f_{55} & \\ & & & & & f_{66} \end{array}\right]$$

In the original theory proposed by Von Mises, strain rate and anisotropy are not considered. The distortional strain energy density can be simplified by Hooke’s law
(4)$$\upsilon_{d}=\frac{1+v}{6E}\left[{\left(\sigma_{1}-\sigma_{2}\right)}^{2}+{\left(\sigma_{2}-\sigma_{3}\right)}^{2}+{\left(\sigma_{3}-\sigma_{1}\right)}^{2}\right]$$

The yield stress of the material under uniaxial loading is expressed as *σ*_s_. Then, the critical distortional energy density can be determined as
(5)$$\upsilon_{\mathrm{ds}}=\frac{1+v}{3E}\sigma_{s}^{2}$$

As the yield behavior is assumed to be governed by the critical distortional energy density, the yield criterion is given by
(6)$$\upsilon_{d}\leq \upsilon_{\mathrm{ds}}$$

For orthotropic materials, the yield criterion can be simplified under uniaxial loading. The uniaxial loading direction (i.e., axis-1) is considered as the principal direction of the material. Then, the coordinate axes of principal stress (i.e., axis-1, 2, 3) coincide with the coordinate axes of the material (i.e., axis-*x*, *y*, *z*), and we have the expression of dilatational strain energy density *υ*_V_ and the strain energy density *υ_ε_* [27]
(7)$$\upsilon_{V}=\frac{1}{18}\sum_{i,j=1}^{3}k_{ij}{\left(1-v_{21}-v_{31}\right)}^{2}\varepsilon_{1}^{2}+\frac{1}{2}\cdot \frac{1}{3^{n+1}}\sum_{i,j=1}^{3}f_{ij}{\left(1-v_{21}-v_{31}\right)}^{n+1}\varepsilon_{1}{\dot{\varepsilon }}_{1}^{n}$$
(8)$$\upsilon_{\varepsilon }=\frac{1}{2}(k_{11}-v_{21}k_{12}-v_{31}k_{13})\varepsilon_{1}^{2}+\frac{1}{2}\left[f_{11}+{\left(-v_{21}\right)}^{n}f_{12}+{\left(-v_{31}\right)}^{n}f_{13}\right]\varepsilon_{1}{\dot{\varepsilon }}_{1}^{n}$$

The distortional strain energy density *υ*_d_ is expressed as
(9)$$\upsilon_{d}=\upsilon_{\varepsilon }-\upsilon_{V}=A_{1}\varepsilon_{1}^{2}+A_{2}\varepsilon_{1}{\dot{\varepsilon }}_{1}^{n}$$
in which *A*_1_ and *A*_2_ are coefficients related to constitutive parameters
(10)$$A_{1}=\frac{1}{2}(k_{11}-v_{21}k_{12}-v_{31}k_{13})-\frac{1}{18}\sum_{i,j=1}^{3}k_{ij}{\left(1-v_{21}-v_{31}\right)}^{2}$$
(11)$$A_{2}=\frac{1}{2}\left[f_{11}+{\left(-v_{21}\right)}^{n}f_{12}+{\left(-v_{31}\right)}^{n}f_{13}\right]-\frac{1}{2}\cdot \frac{1}{3^{n+1}}\sum_{i,j=1}^{3}f_{ij}{\left(1-v_{21}-v_{31}\right)}^{n+1}$$

Once the critical distortional energy density *υ*_ds_ is determined, the yield strain and yield stress can be calculated
(12)$$\varepsilon_{s}=\frac{-A_{2}{\dot{\varepsilon }}_{1}^{n}}{2A_{1}}+\frac{\sqrt{A_{2}^{2}{\dot{\varepsilon }}_{1}^{2n}+4A_{1}\upsilon_{\mathrm{ds}}}}{2A_{1}}$$
(13)$$\sigma_{s}=\frac{B_{1}}{2A_{1}}\left(-A_{2}{\dot{\varepsilon }}_{1}^{n}+\sqrt{A_{2}^{2}{\dot{\varepsilon }}_{1}^{2n}+4A_{1}\upsilon_{\mathrm{ds}}}\right)+B_{2}{\dot{\varepsilon }}_{1}^{n}$$
where *B*_1_ = *k*_11_ − *v*_21_*k*_12_ − *v*_31_*k*_13_ and *B*_2_ = *f*_11_ + (−*v*_21_)*^n^f*_12_ + (−*v*_31_)*^n^f*_13_ are coefficients related to constitutive parameters.

## 4. Yield Criterion Considering Uncertainty

Methods to solve uncertain problems can be categorized into probabilistic, fuzzy, and non-probabilistic methods. The interval method is a typical non-probabilistic method commonly adopted to solve various engineering problems when limited information is available. As the yield behavior is governed by the critical distortional energy density *υ*_ds_ in the proposed criterion, it is assumed that *υ*_ds_ is a type of interval parameter. The expression of $\upsilon_{\mathrm{ds}}^{I}$ can be given as
(14)$$\upsilon_{\mathrm{ds}}^{I}=[\upsilon_{\mathrm{ds}}^{l},\upsilon_{\mathrm{ds}}^{u}]$$
where $\upsilon_{\mathrm{ds}}^{l}$ and $\upsilon_{\mathrm{ds}}^{u}$ are the lower and upper bounds of $\upsilon_{\mathrm{ds}}^{I}$, respectively.

Then, the yield stress considering interval uncertainty is given as
(15)$$\sigma_{s}^{I}=\frac{B_{1}}{2A_{1}}\left(-A_{2}{\dot{\varepsilon }}_{1}^{n}+\sqrt{A_{2}^{2}{\dot{\varepsilon }}_{1}^{2n}+4A_{1}\upsilon_{\mathrm{ds}}^{I}}\right)+B_{2}{\dot{\varepsilon }}_{1}^{n}$$

The solution set for the interval yield stress can be expressed as
(16)$$\Pi =\left\{\sigma_{s}:\sigma_{s}^{I}=\frac{B_{1}}{2A_{1}}\left(-A_{2}{\dot{\varepsilon }}_{1}^{n}+\sqrt{A_{2}^{2}{\dot{\varepsilon }}_{1}^{2n}+4A_{1}\upsilon_{\mathrm{ds}}^{I}}\right)+B_{2}{\dot{\varepsilon }}_{1}^{n},\upsilon_{\mathrm{ds}}^{I}\subset [\upsilon_{\mathrm{ds}}^{l},\upsilon_{\mathrm{ds}}^{u}]\right\}$$

The interval expression of the yield stress $\sigma_{s}^{I}$ can be defined as the smallest closed convex interval solution set
(17)$$\sigma_{s}^{I}=[\sigma_{s}^{l},\sigma_{s}^{u}]$$
where $\sigma_{s}^{l}=\min(\Pi )$ and $\sigma_{s}^{u}=\max(\Pi )$.

The interval vertex method is an effective method to solve interval problems when the functional monotonicity over the investigated ranges is guaranteed. As parameters *A*_1_ and *B*_1_ are positive, the relationship between yield stress and critical distortional energy density is monotonic. Therefore, the exact bounds of output functions can be predicted using the interval vertex method. The lower bound and upper bound of interval yield stress can be expressed as
(18)$$\sigma_{s}^{l}=\frac{B_{1}}{2A_{1}}\left(-A_{2}{\dot{\varepsilon }}_{1}^{n}+\sqrt{A_{2}^{2}{\dot{\varepsilon }}_{1}^{2n}+4A_{1}\upsilon_{\mathrm{ds}}^{l}}\right)+B_{2}{\dot{\varepsilon }}_{1}^{n}$$
(19)$$\sigma_{s}^{u}=\frac{B_{1}}{2A_{1}}\left(-A_{2}{\dot{\varepsilon }}_{1}^{n}+\sqrt{A_{2}^{2}{\dot{\varepsilon }}_{1}^{2n}+4A_{1}\upsilon_{\mathrm{ds}}^{u}}\right)+B_{2}{\dot{\varepsilon }}_{1}^{n}$$

## 5. Results and Discussion

### 5.1. Experimental Results

Figure 3 shows the stress vs. strain curves of the C/SiC composite under different loading speeds (10^−5^ m/s, 1 m/s, 5 m/s). The stress–strain response of the specimen reveals that the material almost behaves linearly in the initial stage; moreover, the curves show a non-linear trend and yield under dynamic loading. It can be observed the stress vs. strain curves rise with the increasing loading speed. The loading speeds of 10^−5^ m/s (quasi-static), 1 m/s, and 5 m/s correspond to the yield stress of 78.7 MPa, 118.8 MPa, 143.5 MPa, the elastic moduli of 80.0 GPa, 102.9 GPa, and 103.2 GPa, and the yield strain of 0.11%, 0.18%, and 0.26%, respectively.

Figure 4 shows the typical failure modes of the woven C/SiC composite under quasi-static and dynamic tensile loading. For quasi-static loading, a crack nucleates from a weak spot and grows into the main crack, following the path with the least resistance (Figure 4a); the crack extension causes stress relaxation and inhibits the nucleation of new cracks, accompanied by the breakage of fiber bundles (Figure 4b). The slow crack propagation results in a fracture surface perpendicular to the loading direction (Figure 4c); eventually, a sudden failure occurs with the growing crack extension. Figure 4d shows the typical fracture morphology under quasi-static loading. The failure surface shows cracks penetrating across the carbon fibers with extensive debonding. Different from quasi-static loading, the applied loading lasts only tens or hundreds of microseconds for dynamic loading conditions. The microcracks are generated with random distribution in the matrix in a significantly shorter time (Figure 4e); subsequently, the cracks randomly outspread to wider regions, accompanied by fiber breakage (Figure 4f); ultimately, the propagation of cracks grows to a main cracking pathway (Figure 4g). A rather rough and extended fracture surface is observed under dynamic loading (Figure 4h). The rough fracture surface indicates the random breakage of carbon fibers. Compared to quasi-loading, fiber bundles’ pullout and breakage play a dominant role in the dynamic failure.

Experimental results for quasi-static and dynamic tension of C/SiC composites at different loading rates are given in Table 1. The applied loading speed ranges from 10^−5^ m/s to 8 m/s, corresponding to the strain rate range from 2 × 10^−5^/s to 99.4/s. The yield stress increases from 72.4 to 170.5 MPa with a relative increment of 135%. The yield strain increases from 0.10% to 0.31%. The strain rate shows a significant influence on the yield behavior of the C/SiC composite. It is attributed to the differences in failure modes under quasi-static and dynamic loading. The microcracks do not have sufficient time for nucleation and propagation under dynamic loading; they thus call for higher stress and more energy to failure [28,29].

### 5.2. Yield Assessment

According to the yield criterion introduced in Section 3, the critical distortional energy density is firstly determined by Equation (9) using the yield stress and yield strain under quasi-static loading. Then, the yield strain and yield stress at different strain rates are determined by Equations (12) and (13), respectively. The experimental results are compared with the yield criterion. The related parameters are shown in Table 2.

Figure 5 shows the comparison of experimental and predicted values of yield stress for C/SiC composites at different strain rates. As shown in Figure 5, the yield criterion agrees with the experimental results. It is observed that yield stress grows with the increasing strain rate, and the variation of yield stress with the log of strain rate shows a sharp upward trend in the higher-strain-rate region. A similar trend is observed for the yield strain of C/SiC composites, as shown in Figure 6. The influence of strain rate on yield strain is observed to be more pronounced than that on yield stress. An increase of 111% in yield stress is found from quasi-static to the highest strain rate of 100 s^−1^, and the yield strain increases from 0.11% to 0.30% with a relative increment of 166%.

For a more reliable yield assessment, the interval uncertainty is taken into account according to the yield criterion introduced in Section 4. The yield stress and strain predicted by the yield criterion considering uncertainty are compared with the experimental results in Figure 7. As shown in Figure 7, the interval bounds are closely related to the uncertain level of critical distortional energy density, which is marked as *χ*. Under quasi-static loading, the experimental results fall within the uncertainty level of 18%, which corresponds to an uncertainty of 9% in the yield stress and yield strain. With increasing strain rate, all test data points fall within an uncertainty level of 45% of the critical distortional energy density, corresponding to an uncertainty of 14% and 11% in the yield stress and yield strain, respectively.

The increase in the uncertainty level of critical distortional energy density is attributed to the different failure modes under quasi-static and dynamic loading. As discussed in Section 5.1, the progressive crack extension results in the final failure for quasi-static conditions, while the uneven fiber bundle breakage accounts for failure under dynamic loading. The random breakage of fiber bundles increases the uncertainty in the absorbed energy of the fracture surface. Thus, the uncertainty level of the critical distortional energy density under dynamic loading is higher than that under quasi-static loading. The agreement with the experimental data shows the validity of the proposed criterion in yield assessment. However, it should be noted that the proposed criterion is an extension of the Von Mises yield criterion. Similar to the Tsai–Hill criterion, the interactions between stresses in different directions are not fully considered. One solution is introducing a new strength tensor to describe the coupling between the normal stresses, as in the Tsai–Wu criterion. The validity of the criterion under multiaxial stress conditions needs further study in future work.

## 6. Conclusions

In this work, the mechanical behavior of woven C/SiC composites under dynamic loading is firstly studied experimentally. Results show that the yield stress, elastic modulus, and yield strain of the C/SiC composite increase with the growing strain rate. The failure mode transitions from progressive crack extension to uneven fiber bundle breakage with the increasing strain rate. Then, a strain-rate-dependent yield criterion is proposed to describe the yield behavior of C/SiC composites under dynamic loading. The predicted results by the yield criterion match well with experimental data. Under quasi-static loading, experimental results fall within the uncertainty level of 18% in the critical distortional energy density, corresponding to an uncertainty of 9% in the yield stress and yield strain, respectively, while, for dynamic loading, experimental results are shown to be within the uncertainty level of 45% in the critical distortional energy density, which corresponds to an uncertainty of 14% and 11% in the yield stress and yield strain, respectively. The uncertainty in the yield stress and yield strain under dynamic loading is higher than that under quasi-static loading due to the different failure modes. The validated yield criterion can be conveniently used to predict the yield behavior of C/SiC composites under dynamic loading conditions.

## Figures and Tables

**Figure 1 materials-15-04679-f001:**
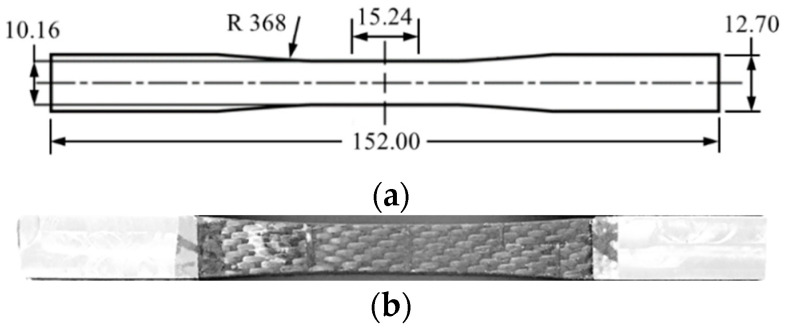
(**a**) Geometry of C/SiC composite specimen (unit: mm). (**b**) Specimen with aluminum alloy sheets.

**Figure 2 materials-15-04679-f002:**
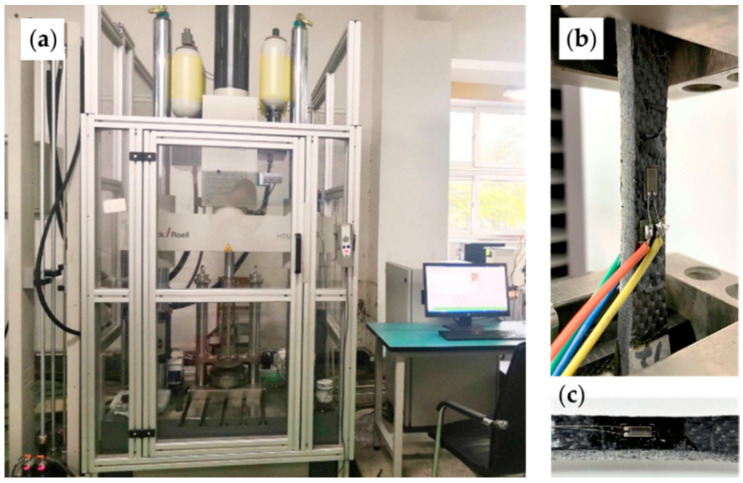
Experimental apparatus: (**a**) HTM 5020 high-speed test machine. (**b**) Grip device. (**c**) Untested specimen and strain gauge.

**Figure 3 materials-15-04679-f003:**
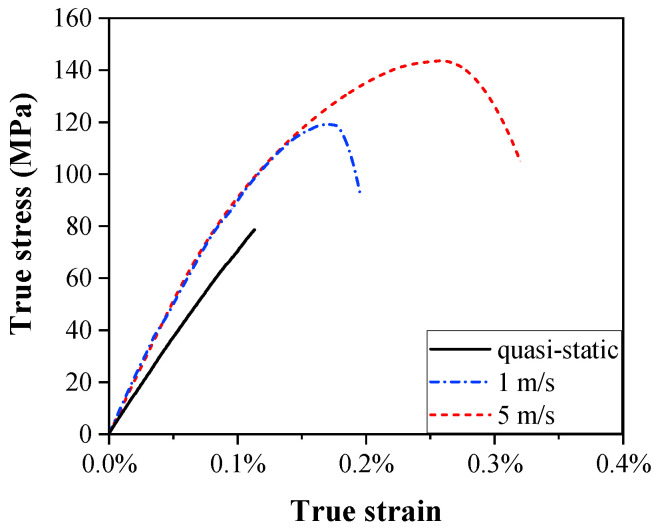
True stress vs. strain curves under different loading speeds.

**Figure 4 materials-15-04679-f004:**
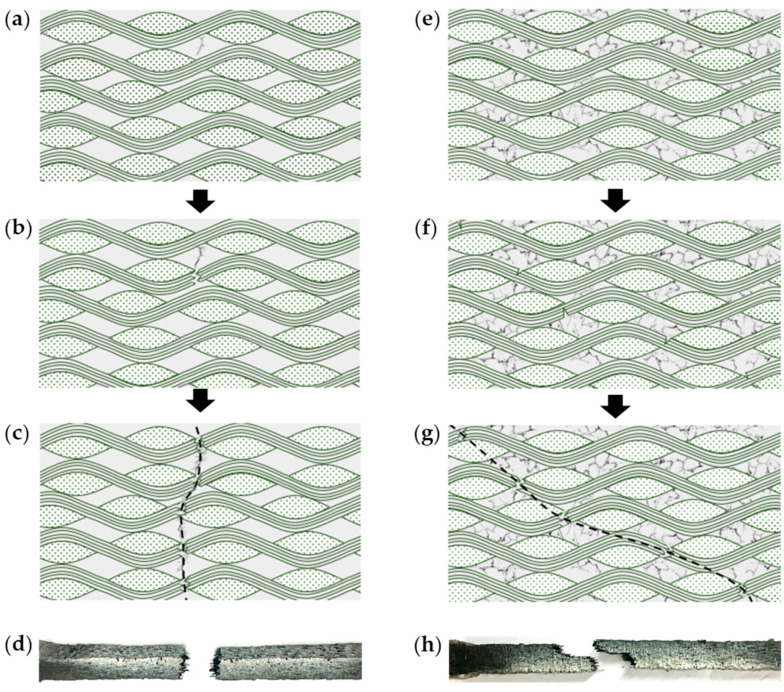
Typical failure mode of the woven C/SiC composite: (**a**) main crack nucleation; (**b**) crack extension; (**c**) cracking path; (**d**) fracture morphology under quasi-static loading; (**e**) multiple crack nucleation; (**f**) fiber bundle cracking; (**g**) fiber bundle breakage and cracking path; (**h**) fracture morphology under dynamic loading.

**Figure 5 materials-15-04679-f005:**
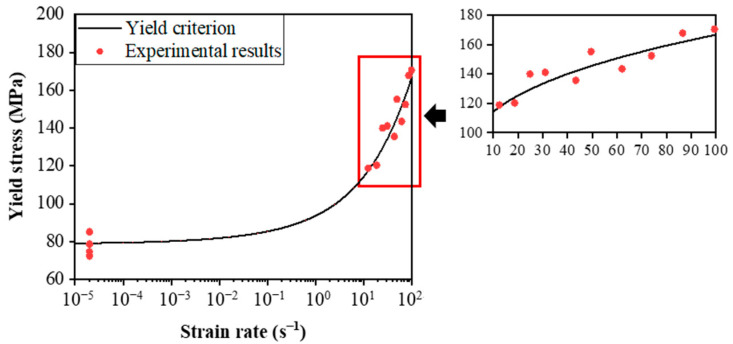
Yield stress vs. strain rate of the C/SiC composite.

**Figure 6 materials-15-04679-f006:**
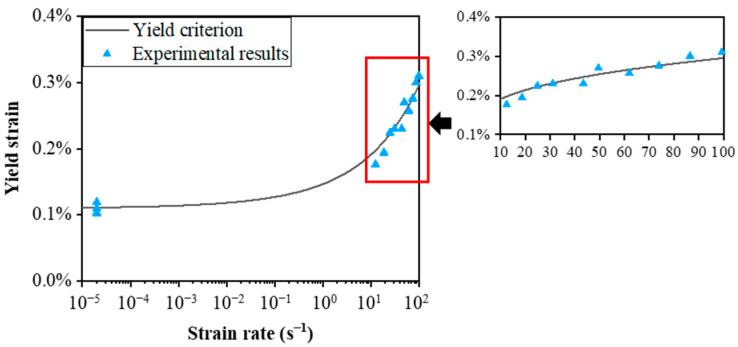
Yield strain vs. strain rate of the C/SiC composite.

**Figure 7 materials-15-04679-f007:**
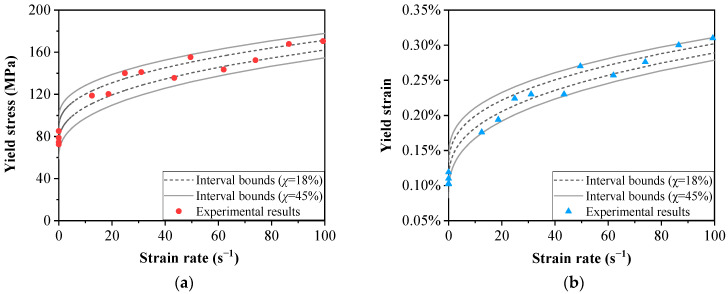
The interval bounds of (**a**) yield stress; (**b**) yield strain at different strain rates.

**Table 1 materials-15-04679-t001:** Experimental results of C/SiC composite under quasi-static and dynamic tensile loading.

Specimen	Cross-Section Area (mm^2^)	Loading Speed (m/s)	Strain Rate (s^−1^)	Yield Stress (MPa)	Yield Strain
S-1	60.83	10^−5^	2 × 10^−5^	78.7	0.11%
S-2	62.41	10^−5^	2 × 10^−5^	85.2	0.12%
S-3	63.37	10^−5^	2 × 10^−5^	72.4	0.10%
S-4	61.45	10^−5^	2 × 10^−5^	74.6	0.10%
D-1	62.01	1.0	12.5	118.8	0.18%
D-2	62.01	1.5	18.7	120.3	0.19%
D-3	63.20	2.0	24.9	140.0	0.22%
D-4	59.99	2.5	31.0	141.1	0.23%
D-5	64.04	3.5	43.4	135.6	0.23%
D-6	60.94	4.0	49.5	155.2	0.27%
D-7	61.25	5.0	62.0	143.5	0.26%
D-8	61.88	6.0	73.9	152.4	0.28%
D-9	58.52	7.0	86.5	167.8	0.30%
D-10	63.05	8.0	99.4	170.5	0.31%

**Table 2 materials-15-04679-t002:** Parameters of the strain-rate-dependent yield criterion.

*υ*_ds_ (MPa)	*n*	*A*_1_ (GPa)	*A*_2_ (GPa·s*^n^*)	*B*_1_ (GPa)	*B*_2_ (GPa·s*^n^*)
3.11 × 10^−2^	0.3	25.69	−1.64 × 10^−2^	71.55	−1.13 × 10^−2^

## Data Availability

Not applicable.

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
