# Peer review of "Dynamic Tensile Behavior of Woven C/SiC Composite: Experiments and Strain-Rate-Dependent Yield Criterion"

_materials, 2022, doi:10.3390/ma15134679_

Round 1
Reviewer 1 Report
1. Authors did not show even a single image of the composite that they have fabricated. Figure 1 is purely a schematic. Add laboratory fabricated composite pictures in the manuscript.
2. Cite the standards utilized in the reference section.
3. English grammar and typos need a significant change/modification.
4. Not enough references are there.
5. Introduction must be broadened to attract more readers.
6. Cite the mentioned latest articles on woven based composites and SiC:
- 2021. Effect of functionalized silicon carbide nano‐particles as additive in cross‐linked PVA based composites for vibration damping application. Journal of Vinyl and Additive Technology, 27(4), pp.920-932.
- 2018. Recent progress in the development of SiC composites for nuclear fusion applications. Journal of Nuclear Materials, 511, pp.544-555.
Reviewer 2 Report
The paper is very interesting and well written as well explained. However, some aspects must be improved to be accepted to be published.
Comment #1:
Experiments
The process of the manufacture of the materials are not well explained. Please include pictures of the process and the properties of each raw material and the final C/SiC composites must be given.
Comment #2:
Figure 2c)
Figure of the strain gauge is not perceptible. Please include a Test sample as Figure 2b) with the strain gauge.
Comment #3:
Figure 4
Please explain each layer of Figure 4. It is not explained the difference between a,b,c and e,f,g
Conclusions:
Quantitatively analysis must be given and not only qualitatively.
Future improved work must be given.

Round 2
Reviewer 1 Report
Accept as is.
Reviewer 2 Report
The request changes were made and accepted by the reviewer.